# Microkinetics of alcohol reforming for $H_2$ production from a FAIR density functional theory database

Qiang Li [1], Rodrigo García-Muelas [1] & Núria López [1]

The large-scale production of hydrogen from biomass under industrial conditions is fundamental for a sustainable future. Here we present a multiscale study of the available reforming technologies based on a density functional theory open database that allows the formulation of linear scaling relationships and microkinetics. The database fulfills the FAIR criteria: findability, accessibility, interoperability and reusability. Moreover, it contains more than 1000 transition states for the decomposition of $C_2$ alcohols on close-packed Cu, Ru, Pd, and Pt surfaces. The microkinetic results for activity, selectivity toward $H_2$, and stability can be directly mapped to experiments, and the catalytic performance is controlled by various types of poisoning. Linear scaling relationships provide valid quantitative results that allow the extrapolation to larger compounds like glycerol. Our database presents a robust roadmap to investigate the complexity of biomass transformations through the use of small fragments as surrogates when investigated under different reaction conditions.

[1] Institute of Chemical Research of Catalonia (ICIQ), The Barcelona Institute of Science and Technology, Avgda. Països Catalans 16, 43007 Tarragona, Spain. Qiang Li and Rodrigo García-Muelas contributed equally to this work. Correspondence and requests for materials should be addressed to N.Lóp. (email: nlopez@iciq.es)

The conversion of biomass to provide chemicals and energy vectors is a fundamental challenge for a sustainable chemical industry based on renewable sources[1,2]. Particularly, to extract hydrogen from biomass, three reforming methodologies as well as direct decomposition (DD) have been put forward (Fig. 1 and Supplementary Table 1). In the steam reforming (SR) process, ethanol, sugars, and other oxygenated compounds react on metals and oxides with steam at temperatures around 400–1100 K to produce $H_2$, CO, $CO_2$, and $CH_4$[3]. Autothermal reforming (ATR) constitutes an improvement over this process as small amounts of oxygen are added along with steam to produce raw syngas. ATR has been tested for ethanol[4] and glycerol[5] on noble metals. However, the high temperatures reached impose limitations to the catalyst thermal stability. Compared with the former, aqueous phase reforming (APR)[6–8], has the advantage of working at temperatures below 650 K, although the process is slow on clean metal surfaces. Only very recently, high turn-over frequencies have been achieved on single-atom Pt/$\alpha$-MoC[9], but long-term stability of this catalyst might be an issue.

A large number of potential catalysts have been experimentally screened for these reactions (Supplementary Table 2). However, large-scale $H_2$ production is still costly. A better understanding of the reaction mechanisms is needed when searching for a high performance catalyst. Theoretical simulations based on density functional theory (DFT) and microkinetic modeling (MK) hold the key for a rational design[10]. However, the most complete kinetic model on alcohol decomposition analyzed only 50 species for ethanol on Pt and about 100 reactions in a correlative global sensitivity analysis[3]. There, the errors inherent to DFT were found to be correlated due to the similar nature of the oxygenated fragments on the metal surface, thus keeping the predictive value of MK based on DFT data. However, the complexity of the compounds derived from biomass has prevented an extensive study of full reaction mechanisms ($C_6$ sugar alcohol decomposition encompasses $10^5$ reactions)[11].

As full mechanistic studies by DFT are non-viable for large alcohols, divide-and-conquer strategies have been put forward, although their representativity has not been fully assessed. The decomposition of small alcohols including methanol, ethanol, ethylene glycol, glycerol, and other oxygenates has been extensively studied. Thermodynamics for the adsorption of intermediates can be obtained from multivariable scaling based on group additivity rules[12–14] and inferred from surrogates. The rate coefficients are then extracted from kinetic-thermodynamic relationships[15–21] derived from key (calculated) decomposition steps either on a single metal, or a small group of alcohols.

However, the computed data is scarce, as only partial networks have been considered and it has been generated with differences in the computational setups (Supplementary Table 3). In consequence, the FAIR[22] (findability, accessibility, interoperability and reusability) nature of the data is not ensured. This prevents the use of large analysis tools to systematize the available information. To diminish the errors, sparsity, and asymmetries in the computational data, here we show a full open database that contains all the decomposition steps of $C_1$–$C_2$ alcohols: methanol, ethanol, ethylene glycol, together with the complementary steps from the water–gas shift reaction (WGSR) and oxygen adsorption. Initially, the database contains the results for the close-packed surfaces, as they are most exposed in the catalytic preparations[23], but can be extended to include lateral interactions, side reactions, other metals, alloys, undercoordinated sites and supports[12,24,25]. In that case, the linear scaling relationships (LSR) previously reported in the literature can also be incorporated[12,24,25]. The database has then been interrogated through microkinetic models to unravel whether the same reaction set is able to reproduce the different experimental behavior on the generation of hydrogen under different reaction conditions, and to predict the best conditions in the reforming of glycerol for one of the metals.

## Results

**Generation of the reaction network**. We have generated a database that can be retrieved from ioChem-BD[26], where we have uploaded the computed 55 reaction intermediates and 215 reactions for the $C_2$ species on each metal (see Data availability section for details). They correspond to 100 dehydrogenations, 55 C–C, and 60 C–O bond cleavages, related to 10, 24, and 21 intermediates in the ethane, ethanol, and ethylene glycol decomposition networks, respectively. Besides, we included 13 reactions to account for the water–gas shift, the $O_2$ decomposition, oxygen-assisted and hydroxyl-assisted dehydrogenation of methanol, ethanol, and ethylene glycol and our previous results on methanol[27]. The procedure to generate the full decomposition network of a given species is presented in Fig. 2. The species can undergo C–C, C–H, O–H, and C–O bond breakings, and each of these reaction products can further experience these four bond cleavages until the formation of the simplest decomposition products: C*, H*, and O* (Supplementary Methods and Supplementary Figs. 14–16). The ground states for all reaction intermediates are described in the Supplementary Discussions. Then, all the adsorption, reaction, and activation energies were computed (Supplementary Tables 4–7). For APR, solvation was included for the adsorption of several molecules (Supplementary Table 5).

Based on the reaction database (Supplementary Table 7), we tested the predictive power of three types of LSR: Brønsted–Evans–Polanyi (BEP), initial state and final state scalings (ISS, FSS). The reactions were classified according to the type of bond breaking, as O–H, C–H, C–C, C–O, and C–OH. For the O–H bond breaking, the BEP slopes, $\alpha$, are between 0.18 and 0.39 (Supplementary Table 8). This indicates that the transition states should resemble the initial states[19]. Indeed, the ISS has lower mean absolute errors (mae) and higher coefficients of determination ($R^2$) than the BEP or FSS relationships. For C–H, C–C, C–O, and C–OH bond breakings, the BEP slopes, $\alpha$,

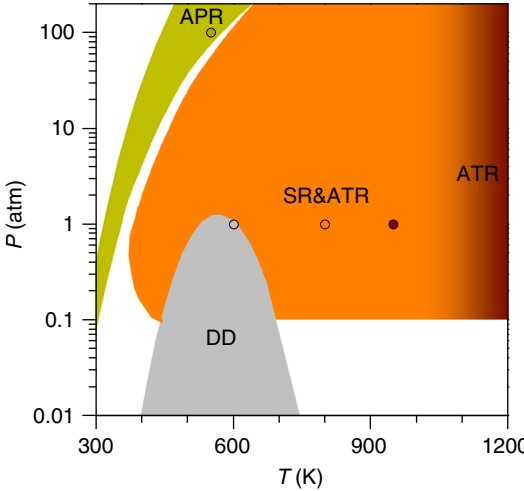

**Fig. 1** Working conditions of reforming processes for hydrogen production. Shaded areas correspond to direct decomposition (DD), steam (SR), autothermal (ATR), and aqueous phase (APR) reforming. The points correspond to the conditions employed in the microkinetic simulations (Supplementary Table 1)

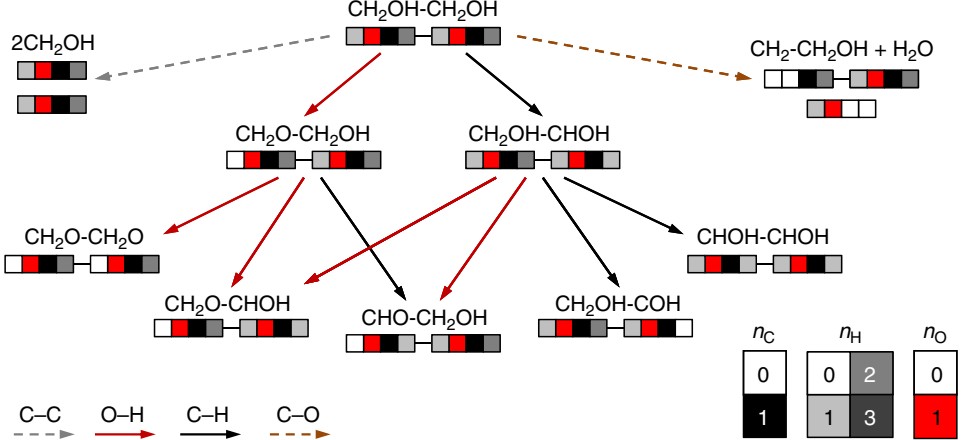

**Fig. 2** Reaction network for the decomposition of ethylene glycol. Each $CH_xOH$ moiety is shown as a quadruplet whose colors are representative of the stoichiometry. The two central boxes are red (O) and black (C) while the first and last boxes indicate the number of hydrogen atoms attached to them, darker gray stands for higher H content. C–C, O–H, C–H, and C–O bond breakings are displayed as gray, red, black, and brown arrows respectively. C–C breakings lead to the lateral path of methanol decomposition, while C–O breakings lead to the lateral path of ethanol and water dehydrogenation. The full set of reactions is shown in Supplementary Figs. 14–16

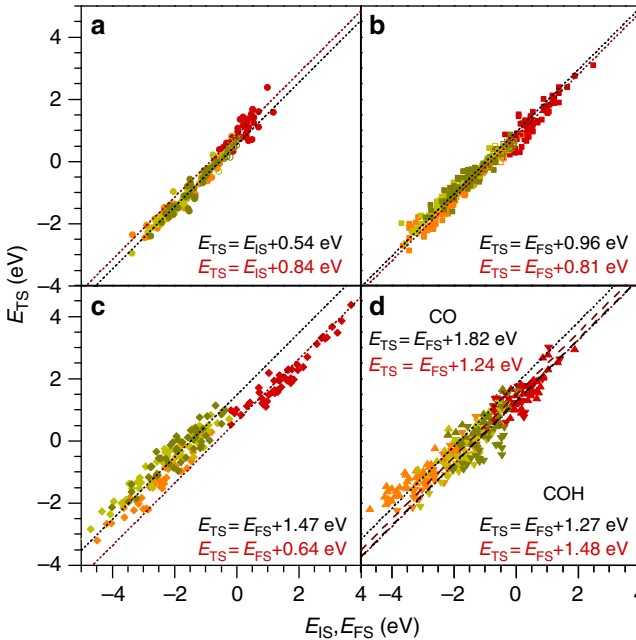

**Fig. 3** Best linear scaling relationships. **a** Initial state scaling relationships for O–H bond breaking on Ru, Pd, and Pt (black dotted line) and Cu (red line). Final state scalings for **b** C–H, **c** C–C, and **d** C–O/C–OH bond breakings. The $\alpha$ (slope) values in the linear scaling relationships were set to 1.00. Transition states calculated with explicit water molecules and/or the implicit solvation model for methanol are taken from ref. [50] and included as empty symbols showing that the LSRs also hold when solvation is considered (Supplementary Note 2 and Supplementary Fig. 1)

are > 0.50, and the FSS are more suitable for the prediction of the transition states energies. The best LSR are presented in Fig. 3. In the ISS or FSS relationships the slope, $\alpha$, was forced to be one (Supplementary Table 9). This leads to a simpler yet representative form for the equations, as it avoids overfitting by eliminating the $\alpha$ regression term. Besides, the LSR equations are independent of the energy reference used (Supplementary Note 1). Both conditions are fundamental to ensure the predictive power of LSR for the activation energies of larger poly-alcohols. Cu follows a

different behavior than Ru, Pd, and Pt, and therefore it was treated independently in Fig. 3. Further details on the linear scalings relationships and the most relevant outliers are provided in the Supplementary Discussions and Supplementary Tables 10–13.

**Pseudo-stationary states in microkinetic modeling**. The complexity of the reaction network can only be described by combining the energy profile with a microkinetic analysis. In Fig. 4a the decomposition of ethanol on Pd is taken as example. To generate this energy profile the transition state with lowest barrier is selected for each intermediate. The rest are only plotted in the figure if their barriers are up to 0.30 eV higher than this reference state. However, in the microkinetic modeling all 252 steps are considered. The interpretation of such a complex profile is not straightforward.

Microkinetics on the DFT results show that a complex reaction network, as the one shown in Fig. 4a, may exhibit several pseudo-stationary states[28], Fig. 4b, c (Supplementary Note 3 and Supplementary Figs. 2 and 3). These pseudo-stationary states were identified to obtain the coverages and hydrogen production reported as crosses in Fig. 5a, b, d, e. Following with the example, in the early stages ($t < 10^0$ s) of ethanol decomposition, Pd dehydrogenates the alcohol moiety, building up a layer of CO* and CH* poisons, Fig. 4b. From $t = 10^1–10^4$ s, C–O breaking starts to be kinetically relevant, leaving CCH$_3$* coverages of 30% while the rest is CO* (66%). Meanwhile, most of the CH* is consumed. The concentration of CO* also increases and reaches a maximum at $t = 10^4$ s (roughly 3 h), while the desorption rate of H$_2$ stabilizes in a plateau value around $1.4 \times 10^{-6}$ s$^{-1}$. The surface behavior for $t = 10^4–10^6$ s would be the most representative pseudo-stationary state. After $5 \times 10^6$ s, the desorption rate of hydrogen is still significant (30% the initial plateau value) and CCH$_2$* and CCH$_3$* cover most of the surface. The final steady state is reached at $t = 6 \times 10^7$ s, roughly two years, and the H$_2$ desorption rate becomes 1% of the previous plateau value. The observed times are only qualitatively meaningful as they have been obtained in a model with no lateral interactions. When those are included, the pseudo-stationary states are the same, both in terms of main products and poisons, but they shifted to shorter reaction times (Supplementary Note 4 and Supplementary Fig. 4). The expandable nature of our database would allow the incorporation of this type of effects, although this discussion is beyond the scope of the present work. Indeed, long equilibration

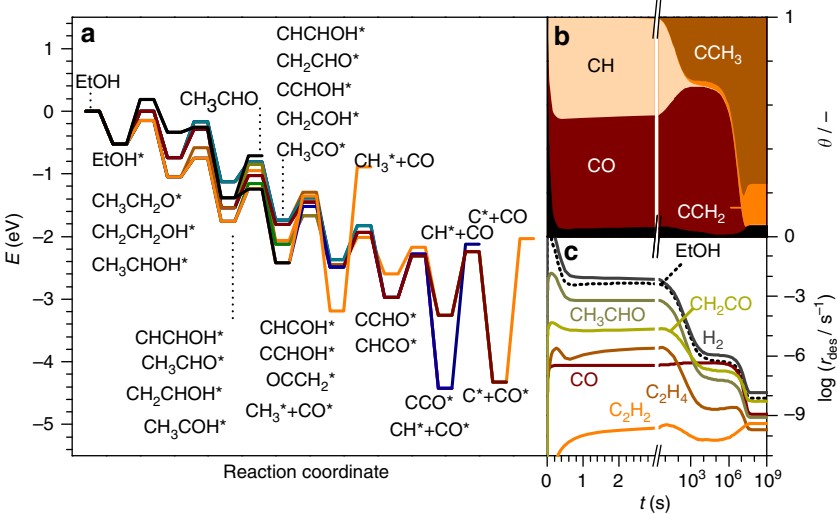

**Fig. 4** Ethanol decomposition on Pd(111). **a** Simplified DFT-Reaction profile, only considering transition states within 0.30 eV energy windows for each TS. **b** Surface coverages as a function of time at the reaction conditions: $T = 600$ K, $P_{ethanol} = 1$ atm. Main intermediates are labeled, while the black area stands for the small intermediate fractions together with the empty sites. **c** Desorption rate of ethanol and several products as a function of time

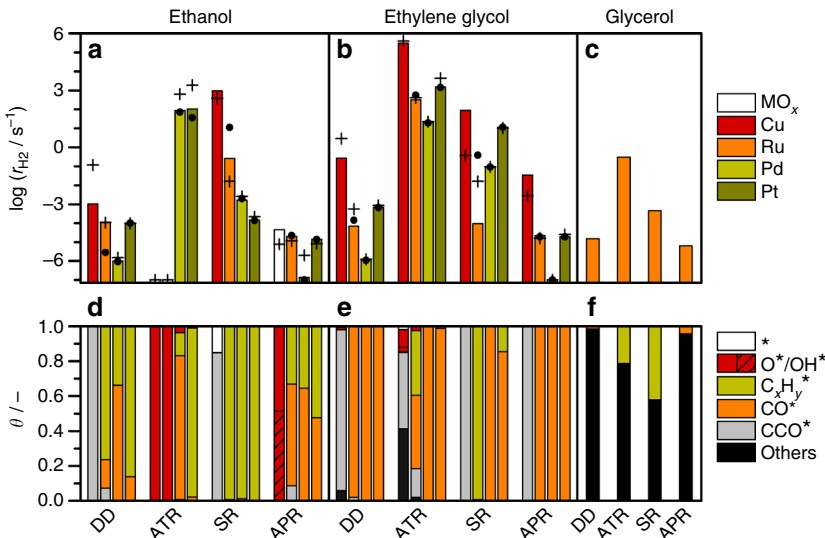

**Fig. 5** Hydrogen production rate during decomposition and reforming. **a** DD, ATR, SR, and APR of ethanol and **b** ethylene glycol on Cu, Ru, Pd, and Pt. **c** Corresponding values for glycerol on Ru. The crosses show the result from MK-DFT, columns from MK-LSR, and dots from the MK-L1O method. White columns stand for reaction conditions where the oxygen content could change the existing catalytic phase. The surface coverages obtained with MK-DFT are shown on panels **d** - **f**

times (in the order of hours) have experimentally been reported for APR of ethylene glycol on Pt[8]. The analysis can then be extended to all metals under all the technical conditions of the reforming. The H₂ production can be found in Fig. 5a, b marked with crosses, along the main surface species (d–e). The analysis is based on the relevant stationary state reached before 3 h. The main adsorbed species and desorption products are listed on Supplementary Tables 14 and 15. Supplementary Discussions provide an analysis purely based on the reaction profiles, which are shown in Supplementary Figs. 6–13.

**Microkinetics of direct decomposition and reforming**. The decomposition of ethanol and ethylene glycol on Cu leaves CCO* as the most abundant intermediate, which covers 99 and 93% of the surface and leaves a small fraction of empty sites, 0.4 and

0.6%. For the ATR and APR of ethanol, the surface coverage of O* and OH* is very high, adding more than 99.9%, thus lowering the productivity of the surface toward H₂. Special care should be taken as suboxides might appear for oxygen coverages higher than 0.75 ML[29], thus compromising the representability of the metal-only model. Acetaldehyde is the main product of ethanol decomposition and reforming, while ethylene glycol yields a mixture of CH₂O, CO, and glyoxal. During ATR, the produced hydrogen reacts with O*, which is in high coverages. This shifts the selectivity toward water and, therefore, no H₂ is produced.

In the ethanol direct decomposition on Ru, the main on-surface species are CCH* (67%), CO* (16%), and CCO* (7%). The main desorption products are CO and H₂, close to the stoichiometric ratio of 2:3. For ethylene glycol, C–C bonds can easily break and the surface is strongly poisoned by CO* (98%) and CCO* (2%). The ATR of ethanol on Ru is a very inefficient

process, and has the lowest hydrogen production rate: less than $10^{-15}$ s$^{-1}$, as the high O* coverage consumes all hydrogen. However, if the oxygen pressure is reduced below 0.09 atm, O* will efficiently remove CO* and the carbonaceous species without poisoning the surface, increasing the H$_2$ production rate to $10^2$ s$^{-1}$ (Supplementary Note 5 and Supplementary Fig. 5). In the steam reforming of ethanol, the main products are CO and CH$_3$CHO, and the whole process occurs at a high rate. In contrast, CO is the most abundant product for the decomposition and reforming of ethylene glycol, along with traces of CO$_2$. On Ru, the highest activities are found for the SR of ethanol and the ATR of ethylene glycol.

The direct decomposition of ethanol and ethylene glycol on Pd is a slow process, as the surface is strongly poisoned by CO* and CCH$_3$*, while producing CO, CH$_3$CHO/HOCH$_2$CHO, and other oxygenates. However, part of the poison can be efficiently removed for the SR and, specially, ATR processes, increasing the yield by 3–8 orders of magnitude. The main products from the ATR of ethanol are CO, CO$_2$, and CH$_3$CHO, while ethylene glycol produces CO, CO$_2$, and glycolaldehyde. In the SR of ethanol and ethylene glycol, the main product is CO, with small traces of C$_2$H$_2$ and CH$_3$CHO. On the ethanol steam reforming, three stable carbonaceous species are formed as poisons: CCH$_3$* (27%), CCH$_2$* (31%), and CCH* (27%), while for ethylene glycol, CO* is the most abundant reaction intermediate. In APR, the main on-surface species are CO* (65%) and CCH$_3$* (27%) for ethanol and CO* (>99.9%) for ethylene glycol. The carbonaceous fragments tend to accumulate rather than desorb.

In the direct decomposition of ethanol on Pt(111), CO is the main desorption product and it is also present on the surface (13%) along CH* (43%), CCH$_2$* (6%), and CCH$_3$* (36%). CO* is the only poison for ethylene glycol, and it is produced along HOCH$_2$CHO close to a 2:1 proportion. The highest H$_2$ productivity is attained for ATR, as the CO* poisoning is efficiently removed by oxidation. The H$_2$:alcohol ratios are 2.5 and 2.6, respectively. For ethanol ATR, the main surface species are all carbonaceous: C* (8%), CH* (62%), CCH$_2$* (11%), and CCH$_3$* (15%), while CO* still blocks most of the sites (98.7%) during for ethylene glycol ATR. In the SR of ethanol, the main desorption products are CO and traces of C$_2$H$_4$ and C$_2$H$_2$, while ethylene glycol yields CO and glyoxal. For ethanol, most of the on-surface species are carbonaceous: CCH$_2$* (80%), CCH$_3$* (11%), CH* (3%), and C* (3%), while for ethylene glycol, CO* is the main poison. Without CO oxidation, the H$_2$ production is up to 8 orders of magnitude slower for SR and APR than for ATR. Nonetheless, a small fraction of water undergoes the WGSR and reduces the CO* poisoning. The H$_2$ productivity is lower for the APR than for SR of ethanol. For the APR of ethanol, CO is the main desorption product, and it is present on the surface along with CH* in roughly 1:1 amounts. Finally, during the APR of ethylene glycol, HOCH$_2$CHO is produced along CO in a 1:5 ratio, but CO is still the main surface intermediate.

**Validation of the microkinetic model**. To assess the robustness of the LSR in reproducing the full MK-DFT data, we built two different microkinetic models, which differ in the methodology followed to obtain the energies: In the first one, MK-LSR, the intermediates energies were taken from DFT, while the activation energies were obtained from the optimum LSR. In the second model, MK-L1O, the activation energies were obtained from LSR but employing a Leave-One-Out procedure for the Ru, Pd and Pt triad. These results are shown as bars and dots in Fig. 5a–c, respectively. The agreement between the simplified methodologies and the full DFT results is thus remarkable, particularly as the relative ordering between the different metals is kept and semi-quantitative values

can be retrieved. The deviations are within two orders of magnitude, within the reliability limits identified by Vlachos[3]. The outliers are the DD of ethanol on Cu, and the SR of ethylene glycol on Cu and Ru. Significant higher activities in the MK-LSR model are found only for ethylene glycol steam reforming on Cu. The production rate of hydrogen is also reasonably obtained with the MK-L1O methodology, showing differences lower than two orders of magnitude with respect to the MK-DFT results. The steam reforming of ethylene glycol on Ru is the only process where MK-L1O deviates by 4 orders of magnitude. The origin of this behavior is that the average activation energies for C–C breakings on Pd and Pt are 0.43 eV higher than the ones from Ru (Supplementary Table 9), so the MK-L1O procedure represents a Ru surface that is much less active than for MK-DFT and MK-LSR. This discrepancy does not appear when taking Pt and Ru, or Pd and Ru, as basic data for the models.

The microkinetic data can be then compared to experimental trends. For instance, the activity for APR of ethylene glycol on silica-supported metal catalyst follows the order Ru>Pt>Pd[8]. This order is well reproduced for MK-LSR, while the MK-DFT and MK-L1O reports Pt>Ru>Pd, being the activity of Pt and Ru almost equal within a 30%. This shows that the rules obtained through LSR might avoid accurate convergence issues in transition state searching, thus providing a slightly more robust framework. Still, intrisic DFT errors would require the improvement of the functionals. On the other hand, the production rate of CO$_2$ follows the order Pt>Ru>Pd[8]. This order is obtained from all three microkinetic models considering together the desorption rates of CO and CO$_2$.

A detailed comparison to the available literature regarding kinetic parameters (rates, apparent activation energies, and reaction orders) has been attempted. Unfortunately, experimental results are sparse and, to the best of our knowledge, none of them compares similar particle sizes, supports, and external conditions for the different materials. In addition, when the catalytic tests are presented together with detailed kinetic analysis, the characterization of the samples before and after reaction is lacking. This is a major handicap when attempting to formulate a robust framework for comparison. However, Table 1 shows a summary of the results in the literature. The microkinetic model has been rerun to replicate experimental conditions[23,30–32]. The systems have been equilibrated to the pseudo-stationary point at $t = 3$ h, then a perturbation, either in the temperature or the pressure, has been applied[33]. The results show that the reaction orders and activation energies are properly reproduced except for the materials containing CeO$_2$, when the support role can be anticipated. The activation barriers show a larger deviation but are qualitative in the same range.

**Microkinetics of glycerol decomposition and reforming**. Finally, the MK-LSR methodology was used to predict the reactivity for glycerol on Ru as, experimentally, it has been a better catalyst in steam reforming than Pd or Pt[34]. The full decomposition network comprises 349 on-surface species and 1950 reaction steps. To this end, the energies for the C$_3$ intermediates were obtained by DFT. Then, the scalings from Fig. 3 were applied to get the activation energies (Supplementary Table 16), and employed in the microkinetic setup. The coverages and hydrogen production can be seen in Fig. 5c, f. The results show that the most suitable technology to produce hydrogen from glycerol is ATR, followed by SR and APR. Experiments indicate that, the longer the alcohol carbon chain (up to C$_6$), the lower the APR selectivity toward H$_2$ and the higher toward hydrocarbons and other compounds[6,7,35]. This is correctly reproduced by our microkinetic model, as we found a H$_2$ desorption rate four times

**Table 1 Comparison between theory and experiments for ethanol SR**

| Ref. | Metal | $T$ | $P_a$ | $P_w$ | $n_a^{exp}$ | $n_a$ | $n_w^{exp}$ | $n_w$ | $E_a^{app,exp}$ | $E_a^{app}$ |
|------|-------|-----|-------|-------|-------------|-------|-------------|-------|-----------------|-------------|
| 30 | Ru/$\gamma$-Al$_2$O$_3$ | 923 | 0.042 | 0.420 | 1.0 | 1.0 | 0.0 | 0.0 | 0.99 | 0.66 |
| 23 | Pd/$\gamma$-Al$_2$O$_3$ | 575 | 0.125 | 0.375 | – | 0.9 | – | 0.0 | 0.49 | 0.54 |
| 23 | Pt/$\gamma$-Al$_2$O$_3$ | 575 | 0.125 | 0.375 | – | 1.1 | – | 0.0 | 0.59 | 0.71 |
| 31 | Pt/$\gamma$-Al$_2$O$_3$ | 575 | 0.125 | 0.375 | 0.8 | 1.1 | – | 0.0 | 0.48 | 0.71 |
| 32 | Pt/CeO$_2$ | 575 | 0.005 | 0.015 | 0.5 | 1.0 | 0.0 | 0.0 | 0.19 | 0.69 |

Apparent activation energies, $E_a^{app}$, in eV
Reaction orders with respect to ethanol and water, $n_a$ and $n_w$ (dimensionless)
$T$: temperature (in K), $P_a$ and $P_w$: partial pressures of ethanol and water (in atm)
The calculations are performed under the same conditions of experiments, labeled "exp"

higher for ethylene glycol than for glycerol. Besides, the selectivity toward H$_2$ was higher for ethylene glycol (84%) than for glycerol (72%). The selectivity reduction of glycerol reforming and decomposition is caused by a plethora of C$_3$H$_y$O$_z$ compounds that populate the surface during all the processes.

## Discussion

We would like to highlight the importance of our present results in the design of new catalysts. First of all, a FAIR[22] database was set up and made accessible to other researchers, allowing its extension to consider lateral interactions, low-coordinated sites, other metals, and supports within the same computational framework. Secondly, the common nature of many elementary steps allows them to be transferred, inherited, and expanded to investigate many reactions on the same metals, thus reducing the computational burden. Third, the completeness in the decomposition path ensures that no intermediates or products are discarded, as it has been done in the literature, where many times only the selective path was identified. This would allow effective machine-learning implementations as those in ref. [36]. Finally, our database can be used as a starting point for increasing our knowledge on BEP and TSS relationships, as well as a training set for machine-learning algorithms related to catalysis. We hope that the standard set by the present approach is adopted by other practitioners in the field to accelerate the comparison of the catalytic properties of different materials and to provide robust design guidelines from modular databases. As physical insights, we would like to highlight that the catalysis phenomena are a function of the reaction conditions, as they control poisoning and thus the state of the catalyst. It is particularly important that as the state of the catalysts defines which routes are more likely within the reaction network. The kinetic data here presented can be further employed in reactor design, to find new conditions for which higher activities and selectivities can be found, as well as to understand the nature of the surface poisons.

In summary, we have investigated the complex reaction networks that arise from the decomposition of ethanol and ethylene glycol on Cu, Ru, Pd, and Pt, through a multiscale method that encompasses microkinetic modeling on the energies obtained through DFT. To this end an open database has been set up to provide the most robust sets of LSR that allow the evaluation of different catalysts. With this scheme, activity, selectivity toward H$_2$, and stability under a variety of technical reforming conditions have been derived and compared to experiments. The present work paves the way for an open, accessible, interoperable, and reusable (FAIR) database for simulations of catalytic properties that can speed up the identification of better performing catalysts in the transformation of biomass compounds.

## Methods

**Computational details**. The density functional theory calculations were performed with the Vienna Ab-initio Simulation Package (VASP)[37,38]. The functional of choice was PBE[39] and the van der Waals (vdW) contributions were obtained through the DFT-D2 method[40,41], with our reparameterization of $C_6$ coefficients for metals[42]. This setup has been proven to predict the experimental adsorption energies of several mono-alcohols and poly-alcohols accurately[14]. The inner electrons were represented by projector-augmented wave pseudopotentials (PAW)[43,44] and the monoelectronic states were expanded in plane waves with a kinetic energy cutoff of 450 eV. Metal surfaces were modeled by a four-layers slab and at least $p$ (3 × 3) supercells, where the two uppermost layers were fully relaxed and the rest fixed to the bulk distances. In the surface calculations, the Brillouin zone was sampled by a $\Gamma$-centered $k$ points mesh from the Monkhorst-Pack method[45], and the $k$ point samplings were denser than 30 Å$^{-1}$. The vacuum between the slabs was at least 13 Å, and the adsorbates were placed only on one side of the slab and thus a dipole correction was applied to remove spurious contributions arising from this asymmetry[46]. The molecules were placed in a cubic box of 20 Å sides. Transition states were located by a combination of the Nudged Elastic Band and the Improved Dimer Method[47–49]. In all cases, the nature of the saddle points was assessed by the diagonalization of the numerical Hessian generated by 0.02 Å displacements for each coordinate. All TS structures have a single imaginary frequency. In all cases the optimization thresholds were $10^{-5}$ eV and 0.02 eV Å$^{-1}$ for electronic and ionic relaxations, respectively. The decomposition reactions can be used to explore all the experimental conditions including water and oxygen effects as: Water–gas shift-related reactions are explicitly included in the reaction pool; Solvation effects[50,51] were included for APR; O-assisted C–H breaking requires 0.60–1.00 eV higher energies than the non-assisted counterparts; OH-assisted proton abstractions in alcohols are roughly barrierless[50]; and O-assisted proton abstractions in alcohols reduce the barriers to $\frac{1}{2}$ the original value (Supplementary Table 7). In addition, the formulations of the catalyst are typically supported on carriers with acid/base characteristics[8] thus, when comparing to experiments with active supports (Table 1), these reactions have been considered to occur on the support and to be barrierless.

**Microkinetic modeling**. The microkinetic model explores hydrogen production on the close-packed surfaces of Cu, Ru, Pd, and Pt under four reaction conditions: DD at constant temperature, ATR, SR, and APR, which are presented in Fig. 1 and Supplementary Table 1. Close-packed surfaces: Cu, Pd, Pt(111), and Ru(0001), were chosen as they are more represented in the equilibrium structure of active metal nanoparticles. In addition, they are smaller than open surfaces and are easier to compare to previous computational data. Moreover, most of the catalyst preparation results in active nanoparticles larger than 5 nm in diameter[23]. Since our database is expandable, it would be possible to add the results from low-coordinated sites, alloys, lateral effects, and carriers[24,25]. The procedure, detailed in the Supplementary Methods and Supplementary Figs. 17 and 18, can be summarized as follows: A stream containing the alcohol, water, and oxygen in variable proportions was fed into an isothermal differential reactor. The reactor operates in transient state and the initial coverages correspond to a clean surface, while the temperatures and pressures resemble typical experimental conditions. The adsorption rates were obtained from the Knudsen equation[33], the rate coefficients from transition state theory[33], and the activation energies from DFT calculations. The mass balance for each species $i$ comes from the sum of rates $j$ in which $i$ participates. A site balance equation was also included. The system of ordinary differential equations was solved in Maple 13. A reaction was considered to reach a relevant stationary state when the surface concentrations and reaction rates varied <0.01% s$^{-1}$.

**Data availability**. The DFT data that support the findings of this study are available in ioChem-BD[26] with the identifier doi:10.19061/iochem-bd-1-37. Instructions about data management are provided in Supplementary Methods. The corresponding labels can be found in Supplementary Figs. 14–16 and Supplementary Tables 7, 16 and 17.

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

## Acknowledgements

We thank the ERC-2010-StG-258406 Bio2chem-d, ERC-2015-PoC-680900 BigData4Cat, MINECO CTQ2012-33826, and MINECO CTQ2015-68770-R projects for financial support. We gratefully acknowledge the generous computing time and assistance provided by the Barcelona Supercomputing Center and the Spanish Supercomputing

Network, BSC-RES. The authors thank Moisés Álvarez-Moreno for technical support in the management of the ioChem-BD database.

## Author contributions

Q.L. and R.G.-M. performed the numerical calculations and contributed equally to this work. N.L. supervised the project. All authors contributed to analyze the data and to write the manuscript.

## Additional information

**Competing interests:** The authors declare no competing financial interests.

