## [Peer Review File · Nature Communications]

this manuscript has been previously reviewed at another journal that is not operating a transparent peer review scheme. This document only contains reviewer comments and rebuttal letters for versions considered at Nature Communications.

Reviewer #1 (Remarks to the Author):

The manuscript impresses with its rigorous implementation of a well-organized and fairly comprehensive database, coupled with the application of linear scaling relationships and microkinetic modeling for a complex reaction network. This work is maybe one of the most complex reaction networks to which the concept of linear scaling has been applied. Thus, it is a convincing demonstration that linear scaling relationships are useful and go far beyond the simpler mechanisms they have been applied to.

While there is no new catalyst or mechanism discovered, and no fundamentally new knowledge generated, the work presents a rather promising approach on how to deal with scaling relationships in the context of complex reaction networks, interchangeability of results from different groups and a systematic extension to even more complex reaction systems. It is fair to say that the presented work constitutes an improvement over the approaches taken in the seminal body of work from Prof. Norskov's group on the topic of linear scaling relationships and catalyst design.

It must be recognized, however, that this paper does not present any groundbreaking or earth-shattering new conclusions; yet, given the popularity of computational catalyst design, a timely publication of this manuscript is recommended, such that current and future efforts in this area can benefit from the lessons learned by these authors.

Reviewer #2 (Remarks to the Author):

Overall, I think the authors have done a reasonable job of responding to the reviews. The most significant issue, which they acknowledge, is that the connection to experimental results is weak (even with the revised Table 1, which honestly doesn't add much), mostly because the number of possible points of connection is very small. For this reason, it's hard to see that the ability to advance catalyst design has been demonstrated, though this work appears to be a step in the right direction, and certainly provides some interesting results. I would therefore support publication in Nature Communications.

Reviewer #3 (Remarks to the Author):

The authors present a multiscale study of alcohol reforming for H_2 production over Cu, Ru, Pd and Pt from DFT database. Enormous amount of energetics and elementary reaction barriers were calculated by vdW corrected DFT, and the microkinetics with help of linear scaling relationship allowed the authors to derive activity, selectivity towards H_2 and stability and extrapolate to large compounds. This presents an important progress of complex biomass related compounds in heterogeneous catalysis and falls well in the audience of the journal. The manuscript is publishable after major revision as indicated below

(1) The reaction network considered focused on alcohol decomposition followed by subsequent water gas shift reaction. Though the reaction conditions for steaming reforming, autothermal reforming and aqueous phase reforming was considered in simulation, the influence of water and oxygen on reaction network was not included in the reaction network. This may affect dramatically the result presented in Figure 5. More careful discussion and elaboration on this should be added in revision.

(2) The multiscale study present valuable information of coverage, overall activity and selectivity etc, this is great ! However, what's the physical insights revealed from the manuscript ? This is particularly important for the manuscript since the comparison between theory and experiment was rather weak.

Answer to Reviewer #1:

The manuscript impresses with its rigorous implementation of a well-organized and fairly comprehensive database, coupled with the application of linear scaling relationships and microkinetic modeling for a complex reaction network. This work is maybe one of the most complex reaction networks to which the concept of linear scaling has been applied. Thus, it is a convincing demonstration that linear scaling relationships are useful and go far beyond the simpler mechanisms they have been applied to.

While there is no new catalyst or mechanism discovered, and no fundamentally new knowledge generated, the work presents a rather promising approach on how to deal with scaling relationships in the context of complex reaction networks, interchangeability of results from different groups and a systematic extension to even more complex reaction systems. It is fair to say that the presented work constitutes an improvement over the approaches taken in the seminal body of work from Prof. Nørskov's group on the topic of linear scaling relationships and catalyst design.

It must be recognized, however, that this paper does not present any groundbreaking or earth-shattering new conclusions; yet, given the popularity of computational catalyst design, a timely publication of this manuscript is recommended, such that current and future efforts in this area can benefit from the lessons learned by these authors.

We thank the Reviewer for his/her positive view on our manuscript. We have improved our discussions to include direct applications in the design of new catalysts and reactors.

Answer to Reviewer #2:

Overall, I think the authors have done a reasonable job of responding to the reviews. The most significant issue, which they acknowledge, is that the connection to experimental results is weak (even with the revised Table 1, which honestly doesn't add much), mostly because the number of possible points of connection is very small. For this reason, it's hard to see that the ability to advance catalyst design has been demonstrated, though this work appears to be a step in the right direction, and certainly provides some interesting results. I would therefore support publication in Nature Communications.

We thank the Reviewer for his/her positive view on our manuscript. As for the impact of our results in catalysts' design our work has two main outcomes. The first one is technical, as we provide a robust database that can be used: (i) as starting point for other simulations (other surfaces, coverage, supports can be included this way); (ii) for increasing our knowledge on BEP and TSS relationships, and (iii) as a training set for machine-learning algorithms related to catalysis. This technical outcome cannot be neglected as it sets up a framework for the integration of data in catalysis. The second outcome is the physical insights. In our view this encompasses: (i) that reaction networks are basically transferrable for different conditions by adding few simple extra chemical equations; (ii) that the catalysis phenomena is a function of the reaction conditions as they control poisoning and thus the state of the catalyst. It is particularly important that as the state of the catalysts defines which routes are more likely within the reaction network. *As an example, we have added the Supplementary Note 3 and Supplementary Figure 10, explaining how the pressure of oxygen controls the catalyst state and the hydrogen production during the ATR of ethanol on Ru. We have also strengthened the discussion of our outcomes on pages 12-13.*

Answer to Reviewer #3:

We would like to thank the Reviewer for his/her positive analysis on our work. In the following we have addressed his/her comments in black, our answers are in blue and the actions taken in italics.

(1) The reaction network considered focused on alcohol decomposition followed by subsequent water gas shift reaction. Though the reaction conditions for steaming reforming, autothermal reforming and aqueous phase reforming was considered in simulation, the influence of water and oxygen on reaction network was not included in the reaction network. This may affect dramatically the result presented in Figure 5. More careful discussion and elaboration on this should be added in revision.

We thank the Reviewer for pointing out this factor that is indeed crucial. Following his/her suggestion, we computed (i) the O–H breakings assisted by coadsorbed O* and OH*, for methanol, ethanol, and ethylene glycol; (ii) the adsorption energies of the major products in liquid environments by using the implicit solvation model developed in our group (*J. Chem. Theory Comput.*, 2016, **12**, 133; *J. Phys. Chem. C*, 2014, **118**, 17531). Our results show that O* and OH* would significantly reduce the barriers for the O–H bond breaking. *These new results have been added to the Supplementary Tables 5 and 7 and the computed structures incorporated to the database. With these new DFT parameters, the microkinetic models were rerun. The final results are discussed on pages 7–9, Figure 5, and Supplementary Tables 14–15. A copy of the old and new versions of Figure 5 are shown below:*

Figure 5 (old).

Figure 5 (new). Hydrogen production rate during the direct decomposition, and reforming: ATR, SR, and APR of (a) ethanol and (b) ethylene glycol on Cu, Ru, Pd, and Pt. (c) corresponding values for glycerol on Ru. The crosses show the result from MK-DFT, columns from MK-LSR, and dots from the MK-L10 method. White columns stand for configurations where the oxygen content could change the existing catalytic phase. The surface coverages obtained with MK-DFT are shown on panels (d-f).

The main difference observed between them is found in ATR operation. While previously only Pt was identified as an active surface, now activity is found also for ethanol ATR on Pd. In the ATR for ethylene glycol, the activity seems to improve for all the systems. However, the appearance of new poisons would require a much more detailed evaluation of the stability of these materials. The other operation modes are only slightly affected by the inclusion of the new reactions.

2) The multiscale study present valuable information of coverage, overall activity and selectivity etc, this is great! However, what's the physical insights revealed from the manuscript? This is particularly important for the manuscript since the comparison between theory and experiment was rather weak.

We agree with the Reviewer that the comparison to experiment is suboptimal, but this is due to the lack of systematic experimental studies. As for the impact of our results in catalysts' design our work has two main outcomes. The first one is technical, as we provide a robust database that can be used: (i) as starting point for other simulations (other surfaces,

coverage, supports can be included this way); (ii) for increasing our knowledge on BEP and TSS relationships, and (iii) as a training set for machine-learning algorithms related to catalysis. This technical outcome cannot be neglected as it sets up a framework for the integration of data in catalysis. The second outcome is the physical insights. In our view this encompasses: (i) that reaction networks are basically transferrable for different conditions by adding few simple extra chemical equations; (ii) that the catalysis phenomena is a function of the reaction conditions as they control poisoning and thus the state of the catalyst. It is particularly important that as the state of the catalysts defines which routes are more likely within the reaction network. *As an example, we have added the Supplementary Note 3 and Supplementary Figure 10, explaining how the pressure of oxygen controls the catalyst state and the hydrogen production during the ATR of ethanol on Ru. We have also strengthened the discussion of our outcomes on pages 12-13.*

Supplementary Figure 10: **Dependence of ATR on the oxygen pressure.** Hydrogen desorption rate for autothermal reforming of ethanol on Ru(0001) as a function of oxygen pressure. $T = 950$ K; $P_{etOH} = 0.25$ atm; $P_{H_2O} = 1.00 - P_{etOH} - P_{O_2}$ [atm]. Yellow and orange regions correspond to first order and zeroth order kinetics. Gray region represents a catalyst poisoned by adsorbed oxygen.

Reviewer #3 (Remarks to the Author):

The revised manuscript addressed properly the main concerns the reviewer raised before, and performed additional calculations for the influence of water and oxygen on the reaction network and kinetics. The authors also acknowledged the weakness of the manuscript, as pointed out by the other reviewers as well. The reviewer have no more comments and recommend the publication as is.

December 20th, 2017

Answer to Reviewer 3.

The revised manuscript addressed properly the main concerns the reviewer raised before, and performed additional calculations for the influence of water and oxygen on the reaction network and kinetics. The authors also acknowledged the weakness of the manuscript, as pointed out by the other reviewers as well. The reviewer have no more comments and recommend the publication as is.

We thank the Reviewer for his/her positive view on our manuscript. We would also like to acknowledge his/her insightful comments which improved our manuscript.

Yours sincerely,
Núria López